# The Future of Community-Based Ecotourism (CBET) in China's Protected Areas: A Consistent Optimal Scenario for Multiple Stakeholders

Bin Zheng [1], Mingchuan Li [2], Boyang Yu [1] and Lan Gao [1,*]

1    Economics & Management College, South China Agricultural University, No.483 Wushan Road, Tianhe District, Guangzhou 510642, China; yukizb@126.com (B.Z.); yuboyangheuet@163.com (B.Y.)
2    School of Business Administration, Nanchang Institute of Technology, No.289 Tianxiang Avenue, Hi-tech Development Zone, Nanchang 330099, China; limingchuan13@163.com
*    Correspondence: gaolan@scau.edu.cn; Tel.: +86-020-85285050

**Abstract:** Community-based ecotourism (CBET) has become a popular strategy to alleviate the contradiction between ecological protection and community development. As the stakeholders of CBET, the community's participation in the planning process is of great importance to in order to realize the sustainability of CBET. Taking a community in Wolong Nature Reserve as a case study, in this study we developed a decision-making participation mechanism based on the participatory scenario method. Through this mechanism, community stakeholders can effectively reach consensus with other stakeholders on the planning of CBET in the future. The results showed that community participation in the planning process can mean decisions are more likely to reflect their interests. They unanimously proposed that future CBET must adhere to the basic principle of protecting biodiversity and must maximize the welfare of the community. Moreover, achieving the sustainability of CBET in protected areas requires the cooperation of all stakeholders.

**Keywords:** protected area; community-based ecotourism; community participation; participatory scenario



## 1. Introduction

In developing countries, the establishment of protected areas is considered to be an effective biodiversity conservation strategy [1]. The establishment of protected areas means the redistribution of resources and rights, which often increases the livelihood vulnerability of the local people [2]. This is because the traditional livelihood strategies of the local people are often highly dependent on the use of natural resources, such as logging, gathering wild non-wood products, hunting, etc., while authorities for protected areas restrict these activities [3]. Strict regulation of natural resource utilization often leads to community hostility towards ecological protection projects, and conflict with the authorities of protected areas [4]. To alleviate this dilemma, authorities may try to implement various market mechanisms to maintain the welfare of local communities. Among them, mechanisms providing local people with sustainable alternative livelihood strategies have been emphasized, particularly ecotourism. Ecotourism, widely recognized as a tool for both biodiversity conservation and community development, is often one of the primary choices for authorities to improve the livelihood of local people. Advocates believe that ecotourism has little impact on natural resources. It can provide sustainable economic benefits to communities, while maintaining ecosystem integrity [5,6]. In fact, more and more studies have found that the effectiveness of ecotourism in protected areas is mixed. Whether ecotourism can become a sustainable livelihood strategy for residents in protected areas has created controversy [7,8]. However, researchers agree that the development of community-based ecotourism (CBET), which promotes local participation in ecotourism planning and management, can guarantee sustainable economic benefits [9,10].

Community participation is one of the keys to the sustainability of CBET. Community participation refers to recognizing local people as important stakeholders in ecotourism development, and giving them control over ecotourism planning and management [11]. Driven by the authorities, more and more locals may participate in the operation of CBET as their livelihood strategy. At present, however, the vast majority of CBET is a mere formality, reflecting a typical "top-down" decision-making process [12]. Some studies have pointed out that the identity of the community as an ecotourism stakeholder is recognized only in the management process [13]. In the process of ecotourism planning, other stakeholders dominate [14]. The community is excluded and becomes a passive recipient of decision-making results, losing the opportunity to put forward their own demands and opinions [15,16]. Due to the deprivation of the right to participate in the planning process, the local community tends to then benefit little from CBET [13]. Therefore, to achieve sustainability, CBET needs to recognize the participation right of locals in the planning process, and improve the ability of locals to effectively participate in this process.

China has made great efforts in biodiversity conservation. By 2017, China had established 2750 nature reserves, accounting for 14.88% of the total land area, exceeding the world average [17]. Because of the highly overlapping geographical distribution with poor areas, these protected areas are facing a strong conflict between protection and development [18]. Authorities hope that CBET can solve this dilemma. More than 70% of nature reserves have developed ecotourism [19]. Most CBET, which emphasizes the community as the main stakeholder, is only nominal, and ignores the issue of locals' participation. In practice, the community has rarely participated in the planning process of CBET. Due to the lack of opportunities and ability to participate in CBET, locals can only passively engage in low-income and unstable businesses [18]. The unfair distribution of ecotourism income often intensifies the contradiction between communities and authorities. It is vital to understand the planning and interest demands of locals for CBET in the future. We need to construct an effective participation mechanism that can enable the community to participate in the planning process of CBET with other stakeholders, and reach an agreement on the future of CBET.

In the study of community participation in the planning and management of regional development, voting methods, multi-standard analysis, and multi-objective linear programming were applied to understand the opinions of locals. The voting method is popular because it is simpler to carry out, and can improve the efficiency of decision-making [20]. Although their operation process is complex [21], hierarchical discrete methods, such as the analytic hierarchy process (AHP) and multi-objective linear programming, are considered to be scientific, comprehensively considering multiple evaluation indexes of the scheme [22,23]. However, these methods fail to take into account the impact of the heterogeneity of participants' behavior preferences on decision-making results.

In this study, based on the participatory scenario method and the triangular fuzzy number multi-attribute decision-making method (TFN-MADM), and considering the behavior preferences of decision makers, we constructed a participation mechanism for joint decision making by multiple stakeholders to explore how to develop CBET in China's protected areas in the future. Our study aimed to (i) determine the key factors affecting the development of CBET; (ii) select the optimal scenario for CBET in the future through the construction and evaluation of different scenarios; and (iii) propose strategies that could be adopted by different stakeholders to achieve a consistent optimal scenario for CBET in the future.

Our study will provide relevant information for CBET in similar protected areas, by demonstrating the importance of community participation in the CBET planning process, where the community can reach an agreement with other stakeholders on the planning of CBET and put forward adaptation strategies and demands. In addition, we constructed a planning participation mechanism for CBET. This will contribute to enhancing the operability and effectiveness of community participation in the CBET planning process,

and achieve the sustainability of CBET by coordinating biodiversity conservation and community development.

## 2. Materials and Methods

### 2.1. Study Area

Since the establishment of nature reserves makes the protection cost to farmers higher than the protection benefit, and at the same time, farmers are also limited in their use of natural resources, the contradiction between the economic development of farmers in nature reserves and local ecological protection leads to differences between farmers in nature reserves and farmers in non-nature reserves. This paper aimed to explore the future of CBET, to enable local residents to create a sustainable development pathway. We chose a village in Wolong Nature Reserve (WNR) as a case study. WNR, founded in 1963 and with an area of 200,000 hectares, is located at the western edge of the Sichuan Basin and southeast of Aba Tibetan and Qiang Autonomous Prefecture in Sichuan Province, and is the oldest and largest giant panda nature reserve in China. It has a forest of 118,000 hectares, accounting for about 56.7% of the total area. Rare and endangered animals and plants are found there, such as giant pandas (*Ailuropoda melanoleuca*), golden monkeys (*Rhinopithecus*), wildebeests (*Budorcas taxicolor*), Davidia involucrata (*Davidia involucrata Baill*), water green trees (*Tetracentron sinense Oliv*), and Sichuan Sequoia (*Larix mastersiana* Rehd. et Wils), which has an irreplaceable role in maintaining the biological integrity and diversity of the whole forest ecosystem. There is a human population of 5343 also in residence. In 1983, Wolong Town and Gengda Town, located in the core area of the reserve, were designated as the Wolong Special Administrative Region of Wenchuan County, under the direct jurisdiction of the reserve administration, to strengthen the protection and management of resources. In the early 1990s, in order to reduce residents' dependence on natural resources and alleviate the contradiction between authorities and communities, the tourism service management organization was established to be fully responsible for the planning and management of ecotourism. With the development of ecotourism, the economy and infrastructure in WNR were improved.

Zumushan village is subordinate to Wolong Town and has a total population of 1062 villagers. This village, with its rich national culture, is inhabited by Tibetans, Qiang, and Han, of which Tibetans and Qiang account for more than 85% of the total population. As of 2017, the per capita annual net income of residents in Zumushan village was RMB 12,900, exceeding the average level of the surrounding areas. CBET has become an important livelihood strategy for residents. There have developed three tourist attractions: Hetaoping, Baoshan pasture, and Lama Temple, as well as two homestay gathering areas, named Moon Bay and Sand Bay. Locals obtain a cash income by providing accommodation, selling tourist souvenirs, and working as tour guides. However, they have no opportunity to participate in the ecotourism planning process, and must passively accept the development model and income distribution scheme of their local ecotourism. Therefore, it is of great significance to promote community participation in the planning process of ecotourism in this area.

### 2.2. Data Collection

In order to fully understand the situation of CBET in the reserve, we held a symposium with relevant personnel in WNR in January 2019, including four reserve managers, five professors in relevant fields, two forestry department managers, two agricultural department managers, two environmental department managers, and two tourism department managers. Subsequently, with the assistance of community staff, we randomly sampled 15 villagers from Zumushan village. We investigated their cognitions regarding and participation in CBET through semi-structured interviews. Of the 15 villagers from different families, two said they were participating in CBET. Of the remaining 13, 9 expressed their intention to participate in CBET in the future. This also shows that the villagers of Zumushan have a strong interest in participating in CBET, but they often fail to convert

their willingness to participate into practical actions due to a lack of funds, lack of labor force, fear of business risks, and lack of business technology. Based on the information collected from the symposium and structured interviews, we, together with the WNR managers, organized a one-day workshop in Zumushan village in June 2019 to discuss the development scenario and achievement strategy of CBET through to 2030. The participants in this workshop included as many different stakeholders of the CBET as possible.

### 2.3. Study Method I: Participatory Scenario

Scenario planning is a method that constructs a variety of possible scenarios in the future based on the key factors driving historical changes. Scenario planning provides a flexible and effective analysis framework for solving uncertain problems in future development [24] (see Figure 1). Based on scenario planning, the participatory scenario strengthens the public's participation in the planning process, and provides an effective decision-making mechanism for stakeholders to reach consensus on future issues [25]. This method was applied to the planning of natural resource utilization [26–28] and the regional development model [29–31]. Usually, the process is to identify stakeholders and then invite them to participate in seminars for scenario planning [32,33].

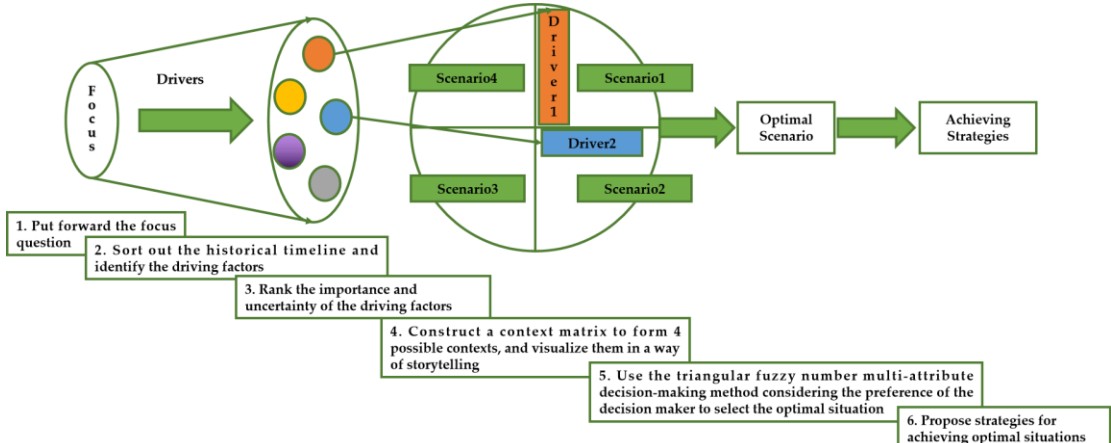

**Figure 1.** The analysis framework of the scenario planning method.

### 2.3.1. Identification of Stakeholders

The construction of future scenarios depends on the ideas of stakeholders who are most familiar with the current situation and will be affected by the decisions. According to the relevant literature [34] and the results of the symposium and semi-structured interview, we propose that the stakeholders of CBET may come from communities, reserve authorities, non-governmental organizations, local governments, and foreign enterprises. Fifteen representatives with different identities were randomly invited to participate in the seminar (see Table 1).

**Table 1.** Identity characteristics of stakeholders in the workshop.

| Stakeholder Categories | Identity Characteristics | Number |
|---|---|---|
| Community | Village cadre | 2 |
| | Homestay operator | 2 |
| | Nature educators | 1 |
| | Residents not participating in CBET | 2 |
| Reserve administration | Reserve manager | 4 |
| NGO | Representative | 2 |
| Local government | Tourism administration manager | 1 |
| Foreign enterprise | Hotel operator | 1 |

### 2.3.2. Identification of Driving Factors

The driving factors used to construct future scenarios had two characteristics: importance and uncertainty. On the one hand, the driving factors will have an important impact on the scenario of CBET in protected areas in 2030. On the other hand, the status of these drivers in 2030 is variable and uncertain. Our study used the historical review method [35] to guide stakeholders to trace the emergence time and major changes in CBET, so as to find a series of factors causing these changes. Then, the stakeholders identified the two most important and uncertain driving factors by ranking them according to importance and uncertainty.

### 2.3.3. Construction of the Future Scenarios

The scenario matrix was constructed based on the interaction between the two most important and uncertain driving factors. Taking the two factors as the coordinate axis, we built a 2 × 2 scenario matrix to form four scenario matrices [36], representing four possibilities of CBET in protected areas in 2030 (See Table 2). In order to make the scenes more vivid, we used story narration to visualize them [25]. Then, we asked stakeholders to discuss the rationality and possibility of each scenario, deepening their understanding of each development mode, and correcting the details of the scenario that were not in line with reality.

**Table 2.** Status of two key driving factors in four scenarios.

|  | Scenario 1 | Scenario 2 | Scenario 3 | Scenario 4 |
|---|---|---|---|---|
| Driving factor 1 | low | high | low | high |
| Driving factor 2 | high | high | low | low |

### 2.3.4. Consistent Choice of Optimal Scenario

The consistent optimal scenario was determined by all stakeholders. Due to different interests, stakeholders' preferences for each scenario may be inconsistent. The optimal scenario of CBET in protected areas needs to take into account the interests of different stakeholders, so the views of all stakeholders should be fully considered in the decision-making process. We asked each seminar participant to evaluate the scenarios according to multiple evaluation attributes, so that they could more accurately express their preferences. At the same time, we used TFN-MADM, considering the behavior preferences of decision makers to calculate the comprehensive evaluation value of each scenario. The scenario with the largest comprehensive evaluation value was determined as the consistent optimal scenario of CBET in the protected areas in 2030. The operation process of TFN-MADM is introduced in detail in Section 2.4.

### 2.3.5. Explore Strategies for Achieving the Optimal Scenario

Finally, the workshop participants compared the difference between the current development of CBET in protected areas and the optimal scenario in 2030, and put forward a series of key strategies to achieve the optimal scenario. In the process of realizing the optimal scenario in 2030, various stakeholders faced different opportunities and challenges, due to different capital endowments. They not only stated the implementation strategies they would be able to implement, but put forward the help they would need, which could be provided by other stakeholders.

### 2.4. Study Method II: TFN-MADM Considering the Behavior Preference of Decision Makers

In a decision-making process, decision makers often make subjective judgments on an abstract scheme according to their intuition or experience, leading to their evaluation results being fuzzy [37]. For example, decision makers often use words such as "good", "average", and "poor" to express their judgments. Triangular fuzzy numbers (TFN) are suitable for describing such fuzzy information [38]. In addition, in reality, decision makers have

different behavior preferences due to the heterogeneity of their own endowments. This will affect the results of decision making, especially for complex and uncertain decisions [39]. TFN-MADM, considering the behavior preference of decision makers, can solve these problems and has been widely used in various fields, such as economic decision making, management science, engineering, and military science [39]. The specific operation process is as follows:

Step one. Select a plan set $U = \{u_1, u_2, u_3, u_4\}$ consisting of four candidate scenarios and an attribute set $S = \{s_1, s_2, s_3, s_4, s_5\}$ consisting of five evaluation indicators, including income, infrastructure, social stability, cultural protection, and ecological environment.

Stakeholders give an overall subjective evaluation score $p_j$ for each scenario $u_j \in U$ ($p_j = \left[ p_j^L, p_j^M, p_j^U \right]$ is a triangular fuzzy number, and $0 \leq p_j^L \leq p_j^M \leq p_j^U \leq 1$), which is the subjective preference value of the overall situation. Then, they give the objective evaluation scores $\widetilde{a}_{ij}$ ($\widetilde{a}_j = \left[ \widetilde{a}_{ij}^L, \widetilde{a}_{ij}^M, \widetilde{a}_{ij}^U \right]$ is a triangular fuzzy number, and $0 \leq \widetilde{a}_{ij}^L \leq \widetilde{a}_{ij}^M \leq \widetilde{a}_{ij}^U \leq 1$) for the evaluation indicators $s_i (i = 1, 2, 3, 4, 5)$ of each scenario $u_j (j = 1, 2, 3, 4)$ to form a decision matrix $A = (a_{ij})_{5 \times 4}$.

Step two. Because the dimensions of different evaluation indexes $s_i$ are inconsistent, it is necessary to transform the fuzzy decision matrix $A = (a_{ij})_{5 \times 4}$ into a standardized decision matrix $R = (r_{ij})_{5 \times 4}$ to eliminate the impact of different physical dimensions on the decision results. Common attribute types are effective benefit type and cost type. The attribute of this study reflected the output of CBET, so the standardized treatment method of benefit attribute was selected. Let $I = \{1, 2, 3, 4, 5\}$, $N = \{1, 2, 3, 4\}$, $r_j = \left[ r_{ij}^L, r_{ij}^M, r_{ij}^U \right]$ and

$$
\begin{cases}
r_{ij}^L = \dfrac{a_{ij}^L}{\sqrt{\sum_{j=1}^n \left( a_{ij}^U \right)^2}} \\[4mm]
r_{ij}^M = \dfrac{a_{ij}^M}{\sqrt{\sum_{j=1}^n \left( a_{ij}^M \right)^2}} \\[4mm]
r_{ij}^U = \dfrac{a_{ij}^U}{\sqrt{\sum_{j=1}^n \left( a_{ij}^L \right)^2}} \\[4mm]
i \in I, j \in N
\end{cases}
\tag{1}
$$

The normalized matrix $R = (r_{ij})_{5 \times 4}$ is obtained. The attribute value $r_{ij}$ is the normalized objective preference value of the scenario $u_j$ of the stakeholders, according to the evaluation index $s_i$.

Step three. Considering that stakeholders may have different behavior preferences, and the difference in decision makers' behavior preferences will affect the decision-making results, we need to set their preferences [40]. We set the behavior preference value of stakeholders as $\lambda$, then the triangular fuzzy number decision matrix $R = (r_{ij})_{5 \times 4}$ can be transformed into a decision matrix with behavior preferences:

$$
F(\lambda) = \begin{bmatrix}
F_{11}(\lambda) & F_{12}(\lambda) & \cdots & F_{1n}(\lambda) \\
F_{21}(\lambda) & F_{22}(\lambda) & \cdots & F_{2n}(\lambda) \\
\vdots & \vdots & \vdots & \vdots \\
F_{m1}(\lambda) & F_{m2}(\lambda) & \cdots & F_{mn}(\lambda)
\end{bmatrix}
\tag{2}
$$

among them,

$$
F_{ij}(\lambda) = \frac{\left[ (1 - \lambda) r_{ij}^L + r_{ij}^M + \lambda r_{ij}^U \right]}{2}
\tag{3}
$$

$F_{ij}(\lambda)$ is the objective preference value of the evaluation index $s_i$ of the scenario $u_j$ when the stakeholders are with the behavior preference $\lambda$.

In the same way, the subjective preference value $p_j = \left[ p_{ij}^L, p_{ij}^M, p_{ij}^U \right]$ can be transformed into the subjective preference value with behavior preference:

$$p_j(\lambda) = \frac{\left[ (1-\lambda)p_j^L + p_j^M + \lambda p_j^U \right]}{2} \tag{4}$$

Step four. Suppose the weight vector of each evaluation index $S = \{s_1, s_2, s_3, s_4, s_5\}$ is $w = \{w_1, w_2, w_3, w_4, w_5\}$, then the comprehensive attribute value of each situation is:

$$Z_j(\lambda) = \sum_{i=1}^{5} \left[ F_{ij}(\lambda)w_i \right], j \in N \tag{5}$$

Due to certain conditions, there is often a certain deviation between the subjective and objective preferences of stakeholders. If the variance $\delta_{ij}^2(\lambda)$ is used to represent the deviation between the objective preference value $F_{ij}(\lambda)$ and the subjective preference value $p_j(\lambda)$ of the evaluation index, that is:

$$\delta_j^2(\lambda) = \sum_{i=1}^{5} \left[ \delta_{ij}(\lambda)w_i \right]^2 \tag{6}$$

Then, the total deviation between the objective preference value $F_{ij}(\lambda)$ and the subjective preference value $p_j(\lambda)$ of all evaluation indicators of the scenario $u_j$ is:

$$\delta_j^2(\lambda) = \sum_{i=1}^{5} \left[ \delta_{ij}(\lambda)w_i \right]^2 \tag{7}$$

In order to make the decision reasonable, the evaluation index weight vector $w$ should minimize the total deviation between the subjective and objective preferences of stakeholders. To this end, the following single-objective optimization model is:

$$min \; \delta(w) = \sum_{j=1}^{4} \delta_j^2(\lambda) = \sum_{i=1}^{5} \sum_{j=1}^{4} \left[ \delta_{ij}(\lambda)w_i \right]^2 \tag{8}$$

$$s.t. \sum_{i=1}^{5} w_i = 1, w_i \geq 0 \tag{9}$$

To solve this model, we set the Lagrange function $L(w, x) = \sum_{i=1}^{5} \sum_{j=1}^{4} \delta_{ij}^2(\lambda)w_i^2 + 2x \left( \sum_{i=1}^{5} w_i - 1 \right)$ and get its partial derivative:

$$\begin{cases} \frac{\partial L}{\partial w_i} = 2 \sum_{j=1}^{4} \delta_{ij}^2(\lambda)w_i + 2x = 0, i \in M \\ \frac{\partial L}{\partial x} = \sum_{i=1}^{5} w_i - 1 = 0 \end{cases} \tag{10}$$

Finally, we obtain:

$$w_i = \frac{1}{\sum_{j=1}^{4} \delta_{ij}^2(\lambda)} \cdot \frac{1}{\sum_{j=1}^{5} \frac{1}{\sum_{j=1}^{4} \delta_{ij}^2(\lambda)}} \tag{11}$$

Substituting the weight value of each evaluation index $w_i$ into formula (5), the comprehensive attribute evaluation value $Z_j(\lambda)$ of the scenario $u_j$ can be obtained.

Step five. According to the different behavioral preferences of stakeholders, we sorted and selected the optimal scenario based on the comprehensive attribute evaluation value $Z_j(\lambda), j \in N$, with the larger the better. The greater the comprehensive attribute evaluation value, the better the scenario.

## 3. Results

### 3.1. Driving Factors of CBET

In general, Zumushan village's CBET has gone through four stages: the planned development stage (1990 to 2003), preliminary development stage (2004 to 2008), depression and decline stage (2009 to 2015), and recovery and development stage (since 2016) (see Figure 2).

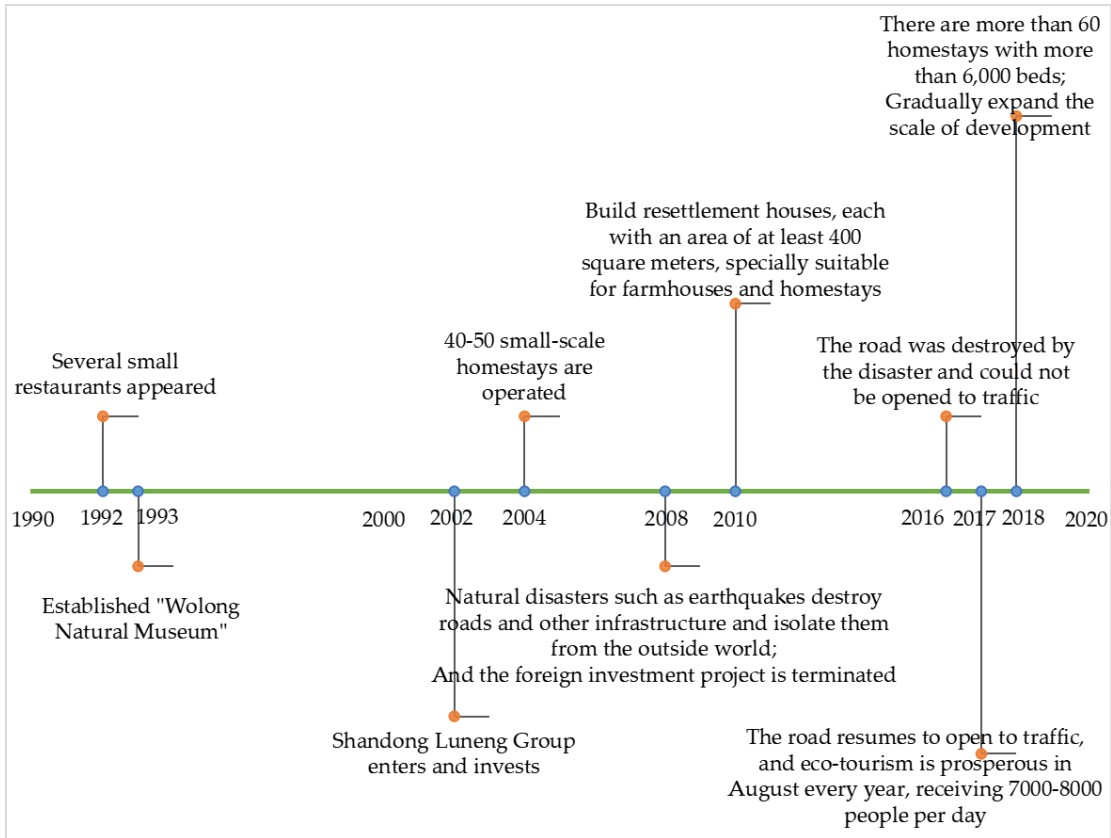

**Figure 2.** The timeline of ecological tourism development in Zumushan Village.

In the 1990, the reserve authority adopted CBET as an alternative livelihood strategy for residents. The authorities built the Wolong Nature Museum, attracting some scientific researchers. Because of the small demand for tourism services and the lack of operating capacity of residents, the local communities did not participate in ecotourism activities. In 2002, investment by the Shandong Luneng Group accelerated the development of CBET. A few local people participated in small-scale businesses and obtained some economic benefits. In 2008, a sudden earthquake and other associated natural disasters destroyed a large amount of infrastructure, such as roads and communications, and the village was isolated from the outside world. Foreign investment projects also withdrew, and CBET almost disappeared. Although the development of ecotourism in Mount Siguniang brought business to a few roadside restaurants during this period, the benefits were very low. In 2016, the post-disaster reconstruction project was successfully completed and the road leading to WNR was restored. The policy support of the authorities and the investment of foreign enterprises have led to the rapid revival of CBET. Moreover, with the help of NGOs, nature education has become a new way for Zumushan village to participate in CBET management. However, compared with CBET in surrounding areas, such as Jiuzhaigou, the tourism projects here are monotonous and seasonal, which makes it difficult to attract tourists continuously and contributes to a lack of profitability.

Based on historical retrospective results, the stakeholders proposed nine driving factors and ranked them according to importance and uncertainty (see Figure 3). The

ranking results showed that infrastructure and entertainment programs are the two most important and uncertain driving factors of the CBET scenario in 2030.

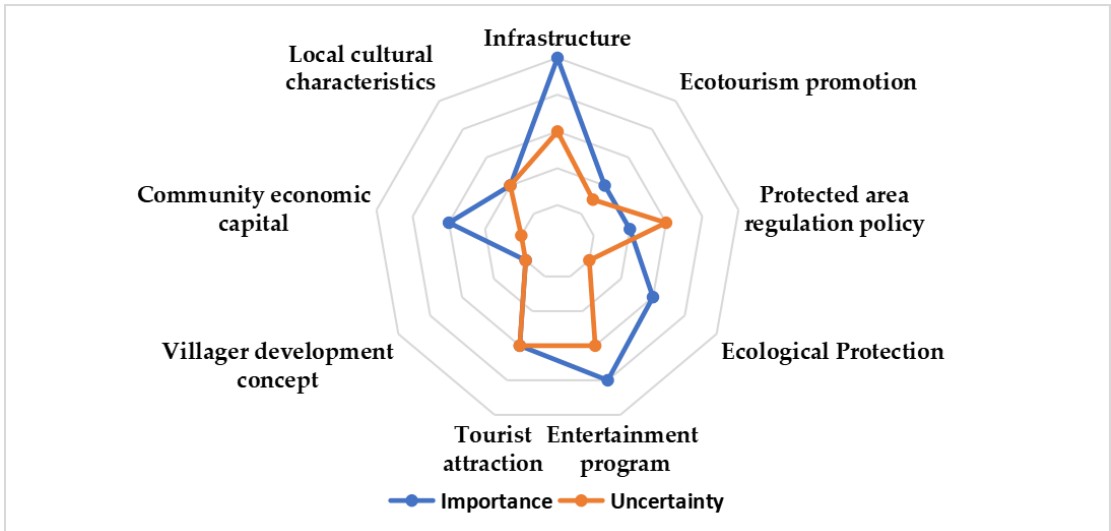

**Figure 3.** Driving factors of ecotourism development and changes in Wolong Nature Reserve in 2030.

Infrastructure is an important material basis for CBET development and community participation in WNR. Historically, the CBET participation and benefits of Zumushan village have been closely related to the improvement of infrastructure. Due to its special geographical location, the region is vulnerable to natural disasters, and the infrastructure faces high natural risks. In addition, the regulation of the reserve administration around the construction of local infrastructure has been strengthened to protect biodiversity.

Entertainment programs are the key factor to attract tourists and drive tourists' consumption, and they are also the main factor affecting the effectiveness of ecological protection. However, the development mode of entertainment programs is controversial. During the discussion, reserve managers and NGO representatives supported the development of environmentally friendly entertainment projects, such as hiking, nature education, etc. Some foreign enterprise managers and residents believe that they need to provide projects with higher economic returns, such as shopping and amusement facilities, to drive local consumption.

*3.2. Four CBET Scenarios in 2030*

Based on two key driving factors, we constructed a scenario matrix (see Figure 4). The horizontal axis of the matrix represents the hardening degree of infrastructure, from weak to strong, and the vertical axis represents the number of entertainment programs, from less to more. Thus, four scenarios were derived, named: "Experiencing Culture", "Nature Paradise", "Get Close To Nature", and "Return To The Original". The names of the scenarios do not reflect the researcher's preference.

3.2.1. Experience Culture

The community retains its original rural appearance and does not add new infrastructure construction. Development of diversified entertainment projects attract tourists and drive tourists' consumption in the local area. Tourists of different ages can meet their own needs here. The middle aged and the elderly can spend the summer here, and enjoy sightseeing, while teenagers can go hiking, experience agricultural work, or learn the customs of ethnic minorities. Residents operate homestays, sell local specialty products, and provide guidance and performances for the tourists.

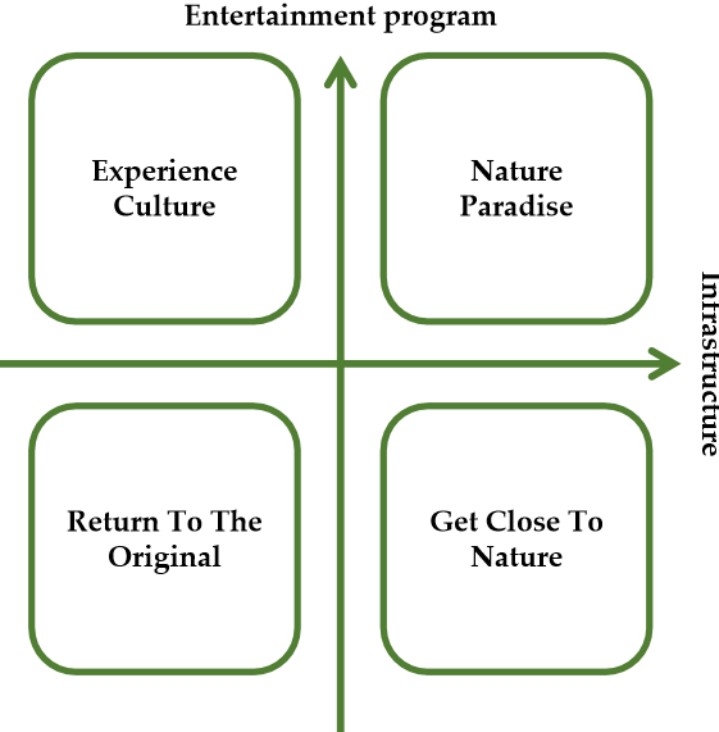

**Figure 4.** 2030 Wolong Nature Reserve Ecotourism Development Situation Matrix.

### 3.2.2. Nature Paradise

Foreign enterprises dominate the management of CBET. They invest capital and build a large number of hardened infrastructures, such as hotels, food courts, shopping plazas, parking lots, garbage recycling sites, entertainment places, etc. They will organize a variety of large-scale outdoor sports (such as horse riding, cycling, etc.), field exploration, nature education, and other activities. In this situation, a few residents operate homestays, while most residents obtain wage income by being employed by foreign enterprises.

### 3.2.3. Get Close to Nature

On the premise of not damaging the ecology, foreign enterprises provide funds to improve the original infrastructure in the village, and build footpaths, parking lots, garbage recycling stations, and science museums. Based on their professionalism and influence in tourism management, foreign enterprises operate and publicize the two projects of forest health and nature education, to provide customized services for small groups of tourists. Residents operate homestays in the form of brand franchises to provide accommodation for these tourists. Alternatively, they can also be employed as staff after participating in training organized by the enterprises. To ensure the interests of the locals, the enterprise withdraws part of the profits as the community development fund.

### 3.2.4. Return to the Original

The community maintains the original rural appearance and does not build new infrastructure, to avoid damaging the biodiversity of the reserve. There are few immigrants in the village, and occasionally a few outsiders come here for scientific research or hiking. Only a very small number of residents provide meals for passing passengers, and the vast majority of residents stay here to engage in agricultural activities, or go out to make a living.

### 3.3. Consistent Optimal Scenario

The participants' subjective preference values for the four scenarios $u_j(j = 1, 2, 3, 4)$ were $p_1 = [0.3, 0.44, 0.6]$, $p_2 = [0.1, 0.33, 0.7]$, $p_3 = [0.4, 0.61, 0.8]$, and $p_4 = [0.1, 0.15, 0.3]$.

Their objective preference values for each evaluation index of each scenario are shown in Table 3.

**Table 3.** Attribute values of four scenarios under five evaluation indicators.

|  | **Experience Culture** | **Nature Paradise** | **Get Close To Nature** | **Return To The Original** |
|---|---|---|---|---|
| Revenue | (0.34, 0.45, 0.56) | (0.52, 0.58, 0.64) | (0.50, 0.57, 0.79) | (0.34, 0.38, 0.44) |
| Infrastructure | (0.37, 0.46, 0.53) | (0.48, 0.55, 0.64) | (0.52, 0.58, 0.79) | (0.34, 0.39, 0.45) |
| Social stability | (0.43, 0.50, 0.58) | (0.41, 0.47, 0.54) | (0.47, 0.52, 0.65) | (0.45, 0.51, 0.58) |
| Cultural protection | (0.48, 0.54, 0.60) | (0.40, 0.46, 0.51) | (0.45, 0.50, 0.62) | (0.43, 0.50, 0.60) |
| Ecological environment | (0.44, 0.49, 0.54) | (0.33, 0.41, 0.49) | (0.49, 0.54, 0.64) | (0.50, 0.55, 0.60) |

Considering the heterogeneity of behavior preferences, the stakeholders had different attribute weight vectors and comprehensive attribute values for each scenario. We set five values of behavior preferences to see whether behavior preferences would affect the decision results. When the preference value was 0.1, the stakeholders were conservative decision makers, while when the preference value was 0.9, the stakeholders were radical decision makers. According to the ranking of the comprehensive attribute values of the four scenarios, we found the optimal scenario for CBET in 2030 (see Table 4).

**Table 4.** Scenario ranking results of stakeholders under different behavior preferences.

| Behavioral Preference | Attribute Weight Vector | Comprehensive Attribute Values for Each Situation | Contextual Ranking |
|---|---|---|---|
| 0.1 | (0.215, 0.237, 0.186, 0.188, 0.175) | (0.454, 0.474, 0.526, 0.438) | $u_3 > u_2 > u_1 > u_4$ |
| 0.3 | (0.228, 0.250, 0.179, 0.178, 0.165) | (0.452, 0.477, 0.528, 0.435) | $u_3 > u_2 > u_1 > u_4$ |
| 0.5 | (0.248, 0.269, 0.168, 0.164, 0.151) | (0.449, 0.482, 0.530, 0.430) | $u_3 > u_2 > u_1 > u_4$ |
| 0.7 | (0.275, 0.292, 0.154, 0.145, 0.134) | (0.445, 0.488, 0.532, 0.423) | $u_3 > u_2 > u_1 > u_4$ |
| 0.9 | (0.312, 0.321, 0.134, 0.121, 0.112) | (0.494, 0.488, 0.533, 0.563) | $u_4 > u_3 > u_1 > u_2$ |

Unless the stakeholders are extremely radical decision makers, the decision results of the optimal scenario of CBET in 2030 were very robust. When the behavior preference value increased from 0.1 to 0.7, the comprehensive attribute values of the four scenarios changed, but their ranking results were fixed. "Get Close To Nature" was always the best scenario, while "Return To The Original" was the worst scenario. The scenario "Get Close To Nature" maximizes the role of CBET in protecting biodiversity and improving community well-being. The scenario "Return To The Original" obviously ignores the needs and interests of community development. Although it has the highest attribute value for the ecological environment, it violates the interests of communities. This means that with regards to future CBET, the stakeholders agreed that the benefits of the community should be maximized alongside the principle of not damaging the ecosystem. It should be noted that when stakeholders have extremely radical behavior preferences (preference value is 0.9), the scenario ranking changes greatly. When the stakeholders are too radical, "Return To The Original" becomes the best scenario. They chose to do their best to maintain the biodiversity of the protected area at the expense of the interests of the community.

*3.4. The Achievement Strategy of the Optimal Scenario*

Excluding the extreme radical behavior preferences of stakeholders, participants considered "Get Close To Nature" as the optimal scenario, and discussed the achievement strategy based on this. Comparing the differences between the current situation of CBET and the optimal scenario in 2030, stakeholders put forward strategies for infrastructure construction, financial capital development, community capacity improvement, community welfare guarantees, and ecological environment protection. At the same time, they identified the specific actions of different stakeholders for each strategy (see Table 5).

**Table 5.** Strategies for achieving the optimal scenario proposed by stakeholders.

|  | Infrastructure Construction | Financial Capital Development | Community Capacity Improvement | Community Welfare Guarantee | Ecological Environment Protection |
|---|---|---|---|---|---|
| Community | Routine maintenance | / | Participate in skill training and broaden your horizons | Pay attention to the issue of equitable distribution in the community | Reduce predatory resource utilization and protect the ecological environment spontaneously |
| Authorities | Provide financial support | Provide preferential policies, such as interest free loans, tax incentives, etc. | Establish a standardized and sustainable training system | Respect the rights and status of communities in CBET. | Strengthen the supervision of ecological environment protection |
| Foreign Enterprises | Introduce advanced environmental protection technology | Invest capital. | Provide skills training | Provide jobs for residents and share CBET benefits in various forms | Develop eco-friendly tourism programs |
| NGO | / | Raise social funds for the reserve and communities. | Provide skills training | Introduce profitable sustainable development programs | Popularize ecological knowledge for the public |

## 4. Discussion

### 4.1. Multiple Stakeholders' Consensus on CBET

Regarding the development of CBET in 2030, community participants finally reached a consensus with other stakeholders. After a debate, all stakeholders unanimously proposed that the future CBET must adhere to the basic principle of protecting biodiversity and maximizing the welfare of communities, which is consistent with the goal of sustainable development of CBET [41–43]. The optimal scenario selected by multiple stakeholders reflected this consensus. This optimal scenario describes that the future development of CBET should reduce the negative impact on the ecological environment, and fully respect the participation of communities and ensure their benefits. Community participation in the planning process ensures the outcomes more closely reflect their interests [9,44]. Even considering the heterogeneity of decision makers' behavior preferences, the consensus is valid unless decision makers have extremely radical behavior preferences.

### 4.2. The Sustainable Development of CBET Needs the Cooperation of Multiple Stakeholders

In addition, the realization of CBET sustainability requires the cooperation of multiple stakeholders [45]. For the community, the sustainable management of CBET has high requirements for capital, technology, and talent, so it urgently needs the support of other stakeholders [46,47]. The development model of CBET, which is dominated by foreign enterprises, participated in by the community, and supported by other stakeholders, was recognized by all participants. Foreign enterprises need to consider the ecological and social benefits of CBET, and invest funds in the fields of ecological diversity protection and community development. As participants in CBET, residents should strive to improve their ability to participate in planning and management, and improve their awareness of ecological protection. Authorities and non-governmental organizations need to provide sustainable support policies and development programs, while monitoring and maintaining the effectiveness of ecological protection.

### 4.3. An Effective Community Participation Mechanism for CBET Planning

The participatory scenario method provides an effective way for communities to participate in CBET planning programs. CBET requires a decision-making process in which multiple stakeholders can discuss and negotiate on an equal platform. However, due to the heterogeneity of their endowments and identities, each stakeholders' cognitions of a problem are biased. In the participatory seminar, the scenario planning method visualizes the issues of the development of CBET in the future, so that stakeholders with different academic backgrounds can equally and fully express their opinions. This is consistent with the results of [29,48,49]. In addition, it provides a platform that enables different stakeholders to listen to the ideas of others. By listening to different voices, each stakeholder can obtain a more comprehensive understanding of CBET's sustainability goals, and they will develop a deeper understanding of each other's positions, which will help to reduce conflict. This is a practice that has been ignored by previous studies related to conflicts between locals and authorities in protected areas [50–52]. They listened to different stakeholders separately, but failed to make stakeholders listen to each other.

## 5. Conclusions

Taking Zumushan village in WNR as a case study, our study obtained some relevant insights for the future development of CBET. The sustainable development of CBET requires the effective participation of local communities in its planning process. Through the participatory scenario method, we identified the driving factors for the development of CBET in the study area and constructed four scenarios. Then, we let each stakeholder evaluate these scenarios to find the optimal scenario, and put forward an achievement strategy for it.

This study suggests that achieving sustainability of CBET in protected areas requires providing a decision-making participation mechanism for different stakeholders, providing them with an opportunity to put forward opinions. Especially for community stakeholders, in addition to participating in the management process of CBET, they need a platform to express their demands. Through this platform, stakeholders can hear different opinions and reach consensus with others on the issues. After discussion, their consensus was to take CBET as a new economic growth point for the community. They hoped to bring economic benefits to the community through CBET, without damaging the local ecology. In addition, they also realized that the development of CBET was inseparable from the cooperation of all stakeholders. Different stakeholders need to make their own contributions to the aspects of infrastructure construction, financial capital development, community capacity improvement, community welfare guarantees, and ecological environment protection, according to their own endowments and capabilities.

Overall, this study also proposes that the participatory scenario method is a powerful tool to achieve sustainability of CBET, because the method enables different stakeholders to express their views and interests equally and fully on CBET, which is conducive to avoiding misunderstandings and conflict. In this way, the right of the community to participate in the planning process of CBET can be guaranteed, and their interests will be heard. This is in line with the original intentions of CBET.

**Author Contributions:** Investigation, M.L. and B.Z.; resources, L.G., methodology, M.L. and B.Z.; writing—original draft preparation, B.Z.; writing—review and editing, B.Z., M.L. and B.Y. All authors were committed to improving this paper and are responsible for the viewpoints mentioned in this work. All authors have read and agreed to the published version of the manuscript.

**Funding:** This research was funded by the National Natural Science Foundation of China, grant number 71761147003 and the National Natural Science Foundation of China, grant number 72003069.

**Institutional Review Board Statement:** Not applicable.

**Informed Consent Statement:** Not applicable.

**Acknowledgments:** We are thankful for the financial support of the National Natural Science Foundation of China and the Consultative Group for International Agricultural Research. We express our appreciation to the anonymous referees and editors of the journal for their constructive comments and suggestions.

**Conflicts of Interest:** The authors declare no conflict of interest.

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
