# Peer review of "The Future of Community-Based Ecotourism (CBET) in China’s Protected Areas: A Consistent Optimal Scenario for Multiple Stakeholders"

_forests, doi:10.3390/f12121753_

Round 1

Reviewer 1 Report

The article focuses on a very important topic of the participation of inhabitants of protected areas in the decision-making process. This is crucial for the durability of the protection. The article describes the situation in an important conservation area of the giant panda. It is well written and interesting. However, it requires a few minor improvements.

L59: It is unclear what "CBET have no real name" means.

L83-109: I would suggest describing the protected area better, especially by giving the total area, land cover types and percentage, and total population (it is not clear now whether the number 1062 refers to Zumushan or Wolong). The Latin names of all the listed animals and plants should be provided.

page 10: Figure 4 do not correspond with later text. For ex. "Get closer to the nature" scenario should be higher on "infrastructure" axis, as it is decribed: "improve the original infrastructure" comparing with "does not build new infrastructure" in "Return to the oryginal" scenario.

page 10-12:The names of the scenarios should be standardized, also in terms of the spelling of capital letters.

page 13-14: The final paragraphs should be completed.

Reviewer 2 Report

Dear authors:

In my opinion the paper fits the scope of the Forests journal well and you have studied a very interesting and practical topic. The manuscript is well structured and wrote, but, in my opinion, some changes should be done before I can recommend the publication.

Introduction:

The framework should be completed emphasizing the problem of the human-wildlife conflicts and the relevance of the ecotourism to promote as a sustainable alternative to the local development. Moreover, it should be added some proposal/applications of several methods to address the relationship between local communities-wildlife conservation. As example, the AHP, Linear Programming or voting methods have been used to this goal.

Here you can find a reference to help you finding some elements/references for the suggested framework:

de Castro-Pardo, M., Pérez-Rodríguez, F., Martín-Martín, J. M., & Azevedo, J. C. (2019). Planning for Democracy in Protected Rural Areas: Application of a Voting Method in a Spanish-Portuguese Reserve. Land8(10), 145.

Material and methods:

The study area should be more-in-depth described, addressing the identified problems regarding the conservation of the Giant Panda.

An example of the semi-structured interviews should be added as Supplementary material.

The Scenarios should be more in depth explained. I encourage the authors present a table with the description and the elements included in each scenario.

What is the propose of the use of this method? Why fuzzy functions? These questions should be explained in the manuscript and the methods should be justified.

Discussion: The Discussion section presents the main results since a very practical approach. The discussion should be more-in-depth completed with some type of theoretical discussion and the results should be compared with other similar studies and discussed.

Minor changes:

2.3.1. Please, review the tittle of this section.

Line 142. Delete the point after ( see Table 1.)

4.1. Please, check the capital letters.

Rewrite the sentence: “This study believes…” (line 418)

Round 2

Reviewer 2 Report

Dear authors,

The requested changes have been completed and the manuscript has been improved enough.